# CycleIE: Robust Document Information Extraction through Iterative Verification and Refinement

## Abstract

In document AI, reliable analytics require converting long, noisy (often multi-document) corpora into *heterogeneous structured data—e.g.,* **tables** for numerical fields, **graphs** for entity–relation structures, **trees** for hierarchies, and faithful **text chunks**. Yet one-pass LLM extraction often yields incomplete or inconsistent structures because it lacks explicit verification and opportunities to revise earlier choices. We present CycleIE, an iterative information extraction (IE) framework that *closes the loop* between reasoning and acting by coupling ReAct with Monte Carlo Tree Search (MCTS). CycleIE employs a multi-agent workflow orchestrated through ReAct and optimized via MCTS to iteratively retrieve, structure, extract, and refine extracted content under verification guidance. This design treats extraction as a search process with feedback, enabling systematic correction of omissions and inconsistencies that defeat one-pass methods, and remains orthogonal to retrieval-augmented generation (RAG) by operating directly over user-provided documents. Experiments on challenging the document-based QA benchmark demonstrate that CycleIE delivers >**10% relative improvements** in extraction quality over strong one-pass baselines, with the largest gains in lengthy or multi-document contexts.

## 1 Introduction

Large Language Models (LLMs) are powerful tools for document AI, yet reliable analytics over long and noisy corpora ultimately hinges on turning unstructured text into *heterogeneous structured data* that downstream methods can directly consume (*e.g.,* tables, graphs, trees, and text chunks) (Wan et al., 2024; Zhu et al., 2023). Directly prompting LLMs to answer analytical queries without such structured extraction often yields vague, incomplete, or inaccurate outputs, especially on lengthy or multi-document contexts (Figure 1(a–b)). In particular, Figure 1(b) highlights superficial reasoning driven by limited long-context handling (Li et al., 2024; Wang et al., 2024b) and the inability to aggregate dispersed evidence (Zhang et al., 2024a; Ling et al., 2023). These observations motivate a structure-first approach that carefully extracts the right format for the task before reasoning.

In practice, the effectiveness of analytical applications fundamentally depends on accurate, **task-specific information extraction (IE)** from complex documents (Dagdelen et al., 2024; Appalaraju et al., 2021). This need spans finance (*e.g.,* extracting key metrics from quarterly statements), law (*e.g.,* compiling evidence across filings), and healthcare (*e.g.,* summarizing patient records) (Kong et al., 2024; Nie et al., 2024; Ntinopoulos et al., 2025). Across these domains, the quality of downstream insights is largely determined by how effectively unstructured text is transformed into *the appropriate structured representation* for the target analysis (Li et al., 2025; Fang et al., 2024).

**Challenges of Task-Specific IE.** Accurately extracting heterogeneous structures for specific analytics tasks requires synthesizing information dispersed over long and multi-source documents while choosing the right structure for the question. This often leads to incomplete extractions, inconsistent entities, or irrelevant content being included. These difficulties are amplified in the presence of complex layouts and long-range dependencies.

**Limitations of One-Pass IE.** State-of-the-art IE pipelines frequently adopt a **one-pass** strategy, directly prompting an LLM to produce the structured output in a single step (Liao et al., 2024; Wang et al., 2024a). As illustrated in Figure 1(c), even strong LLMs lack explicit verification and

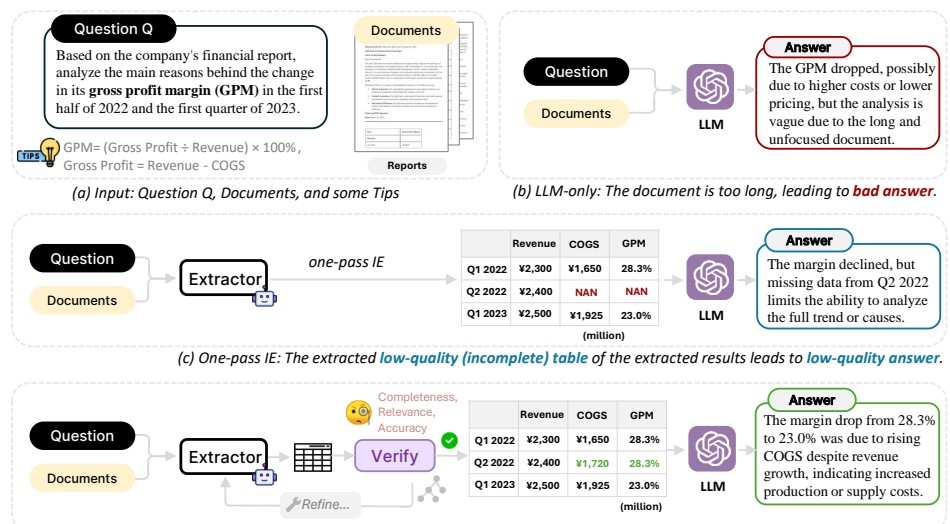

*(a) Input: Question Q, Documents, and some Tips*

*(b) LLM-only: The document is too long, leading to **bad answer**.*

*(c) One-pass IE: The extracted **low-quality (incomplete) table** of the extracted results leads to **low-quality answer**.*

*(d) Iterative IE: By iteratively verifying the extracted information, **high-quality table** was obtained, leading to a **high-quality answer**.*

Figure 1: (a) Input question $Q$ over given documents. (b) Without IE + LLMs for QA. (c) One-Pass IE + LLMs for QA. (d) Iterative IE + LLMs for QA.

opportunities to revise earlier choices (structure selection, extraction scope, and values) within a single pass. Consequently, one-pass IE often yields incomplete or inaccurate structures, underscoring the need to move *beyond* direct, single-shot extraction (Xu et al., 2024; Madaan et al., 2023).

**From One-Pass to Closed-Loop IE.** To address these limitations, we advocate a **closed-loop (iterative)** IE paradigm (Figure 1(d)): verification signals (completeness, relevance, accuracy) guide whether to re-retrieve, re-select structure, or re-extract, and the system repeats this loop until the structured output is auditable and sufficient for downstream reasoning. This perspective explicitly *decouples* structured extraction from reasoning, operates over user-provided documents (orthogonal to retrieval augmentation), and treats extraction as a verification-guided search over actions rather than a one-shot generation. We instantiate this paradigm with CycleIE, which couples ReAct with Monte Carlo Tree Search (MCTS) to select the next action under verification feedback and employs an anomaly-intervention mechanism to avoid verification deadlocks.

**Contributions.** Our contributions in this work are threefold:

(1) **Closed-Loop Iterative IE Framework.** We propose CycleIE, a multi-agent, verification-driven *closed-loop* IE framework for heterogeneous structured extraction in document AI. Orchestrated via ReAct, CycleIE explicitly decouples structured extraction from downstream reasoning and converts long, noisy document segments into auditable structures for reliable analytics.

(2) **Verification-Guided Action Selection.** We integrate ReAct with MCTS to formulate extraction as a *verification-guided search* over actions (retrieve, extract, verify, refine). Explicit verify signals guide action choice, while intervention mechanisms avoid deadlocks—yielding verifiable short-term gains in completeness and accuracy.

(3) **Extensive Experiments and Findings.** On the *Loong* benchmark (10K–250K tokens), CycleIE surpasses the state of the art (StructRAG) by **+9.3–15.3** in LLM scores and **+8–16** in EM. Compared with direct Qwen2-72B prompting, it achieves up to **2.31×** higher LLM scores and **4.33×** higher EM, with only a **15.3%** drop as context grows. These results confirm that moving from one-pass to closed-loop extraction yields robust, high-fidelity structures for long-document analytics.

## 2 PRELIMINARY

**Task-Specific Information Extraction (IE).** In *document AI*, a data analytics task $T$ operates over a user-provided collection of documents $D = \{d_j\}$. The goal is to extract the *heterogeneous structured data* needed to reliably solve $T$ before performing any downstream reasoning.

**Unit Questions.** A complex task $T$ can be decomposed (optionally by a planner) into unit questions $\{Q_1, \ldots, Q_n\}$, where each $Q_i$ targets a *single* structured form (*e.g.,* a table of metrics, a relation graph, a hierarchy, or faithful text snippets). For each $Q_i$, CycleIE iteratively (i) retrieves relevant segments $R_i \subseteq D$, (ii) selects an appropriate structure, (iii) extracts a candidate structure $S_i$, and (iv) verifies whether $S_i$ is complete, relevant, and accurate enough to support downstream reasoning.

**Structured forms.** We consider four canonical structures that cover most analysis needs:

- *Table* (⊞): Organizes data into tables for representing categorical or numerical information.
- *Graph* (◐): Represents entities and their relations as nodes and directed/undirected edges.
- *Tree* (⛁): A hierarchical structure with a root and branches, suitable for nested relationships.
- *Text Chunks* (≡): Coherent segments of text preserving semantic meaning for further analysis.

**Evaluation protocol.** Because ground-truth structures are often unavailable, we evaluate the **end-to-end** effect of structured extraction by comparing **without IE:** $\mathsf{LLM}(T, D)$ (direct reasoning over raw documents) and **with IE:** $\mathsf{LLM}(T, E_{T,D})$ (reasoning over CycleIE-produced structures $E_{T,D}$; if $T$ is decomposed, CycleIE runs per $Q_i$ and aggregates).

Final answers are scored against the **ground truth of** $T$. This isolates the contribution of heterogeneous structured extraction to task performance.

*Remarks.* (a) *No document retrieval.* CycleIE operates on user-provided $D$; it is orthogonal to RAG that retrieves external documents. (b) *End-to-end comparison.* For decomposed tasks $\{Q_1, \ldots, Q_n\}$ (*e.g.,* via CoT or DeepSeek-R1), we compare final answers under the same reasoning model. (c) *Extensibility.* Additional formats (*e.g.,* JSON, XML, domain schemas) can be supported without changing the loop.

## 3 THE CYCLEIE FRAMEWORK

CycleIE employs a multi-agent framework, where specialized **Agents** perform distinct **Actions** and communicate via structured **Signals** carrying feedback and intermediate results. These components are orchestrated through the **ReAct** paradigm, enabling adaptive and context-aware processing. To guide action selection, CycleIE integrates **Monte Carlo Tree Search (MCTS)** for evaluating and refining action sequences.

Given a complex analytical task $T$, CycleIE decomposes it into unit questions $\{Q_1, Q_2, \ldots, Q_n\}$, each triggering iterative information extraction and reasoning. Extracted data is structured and fed into an LLM for sub-question answering, whose intermediate reasoning further informs the next steps. The following subsections detail the Agents (Section 3.1), ReAct-based orchestration (Section 3.2), and MCTS-based optimization (Section 3.3), for iterative IE of a unit question.

### 3.1 THE AGENTS OF CYCLEIE

During the verification process, the system identifies potential causes of poor extraction quality. These may include incomplete information from retrieval that hinders reasoning, missing data detected during extraction, or unit questions that fail to retrieve relevant information and thus require refinement. We define these causes as **Signals**—informative cues that can be passed along in the reasoning loop to guide the optimization of retrieval results, improve extraction quality, or refine the unit questions themselves. Specifically, CycleIE employs six specialized **Agents** (illustrated in Figure 2(a)) to iteratively process each unit question $Q_i$:

**Retriever.** Initially identifies and retrieves relevant document segments $R_i$ to address $Q_i$. It also receives explicit verification signals from the **Verifier**, prompting further retrieval iterations if initial segments are incomplete or inadequate.

**Structurer.** Determines, in a single execution, the optimal structural representation (*e.g.,* table, graph, tree, or text chunks) suitable for effectively answering $Q_i$.

**Extractor.** Converts retrieved segments $R_i$ into structured data $S_i$, adhering to the format selected by the **Structurer**. Upon receiving verification signals indicating issues, it iteratively adjusts extraction processes and refines the extracted data.

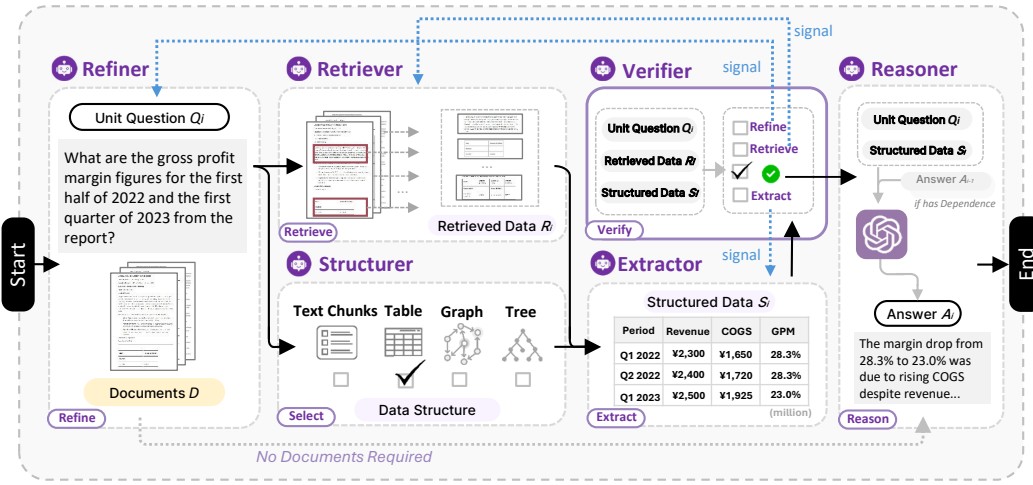

(a) Workflow for processing an individual unit question.

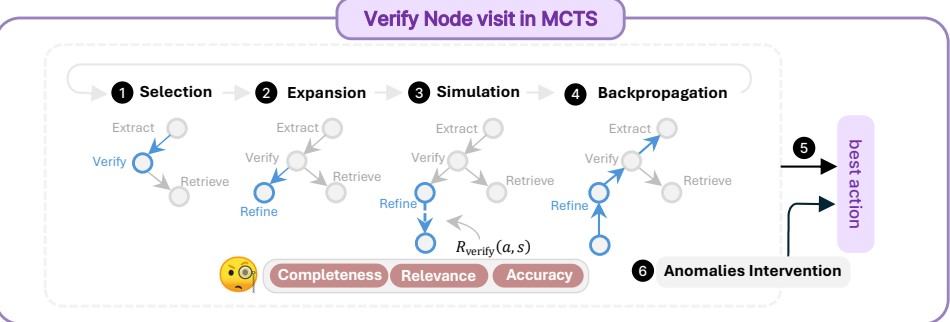

(b) Illustration of MCTS decision-making when reaching the **'Verify'** node.

Figure 2: Overview of CycleIE for one unit question.

**Verifier.** Evaluates structured data $S_i$ produced by the **Extractor** for completeness, relevance, and accuracy, generating explicit verification signals that guide iterative refinement by the **Retriever, Extractor, and Refiner**.

**Refiner.** Responds to verification signals from the **Verifier** to iteratively adjust queries, in order to improve extraction quality.

**Reasoner.** Executes once per unit question, utilizing the final verified structured data $S_i$ to derive clear and interpretable intermediate answers $A_i$, guiding subsequent analytical processes $Q_{i+1}$.

For more detailed technical descriptions of each agent, see Appendix A.

### 3.2 ORCHESTRATING AGENTS VIA REACT

CycleIE orchestrates six specialized agents using the ReAct paradigm, enabling explicit inter-agent communication through **Actions** and **Signals** between agents. The reasoning state $s$ maintains a unit question, retrieved segments, structured data, and verification feedback. Actions are executed by agents to advance extraction and reasoning, while Signals guide coordination and iterative refinement. Table 1 summarizes the six core actions and corresponding signals.

### 3.3 ACTION SELECTION WITH MCTS

CycleIE employs Monte Carlo Tree Search (MCTS) to optimize agent action sequences beyond the reactive bounds of ReAct. The decision process is framed as a tree-structured search problem, where each **node** $v$ represents an **action** $a$ taken in a particular **state** $s$, *i.e.,* the current reasoning context as defined in Section 3.2. Each edge encodes a transition from one action-state pair $(s, a)$ to another $(s', a')$, guided by transition rules $\mathcal{T}(a)$ that define which actions are available after the current action—for example, the *verify* action is only valid after an *extract* action has populated relevant content. A **path** in the tree represents a full action sequence toward a terminal reasoning state.

Table 1: Core Actions and Verifier-Driven Signals in CycleIE

| Action 🔧 | Agent 🤖 | Purpose / Trigger |
|---|---|---|
| **refine** | Refiner | Improve unclear questions; triggered by Verifier on ambiguity |
| **retrieve** | Retriever | Re-fetch better segments; triggered by Verifier on poor retrieval |
| **select** | Structurer | Choose optimal structure for question representation |
| **extract** | Extractor | Extract structured content from retrieved segments |
| **verify** | Verifier | Assess data quality; triggers new actions via signals |
| **reason** | Reasoner | Conduct final reasoning based on structured data |

| Signal 📋 | Agents 🤖→🤖 | Content |
|---|---|---|
| **Poor question** | Verifier→Refiner | Request reformulation or decomposition |
| **Poor retrieval** | Verifier→Retriever | Request keyword revision and rerun retrieval |
| **Poor extraction** | Verifier→Extractor | Request extract suggestion and rerun extract |

MCTS explores and simulates such paths, selecting the next action by balancing local verification signals and long-term reasoning rewards. To prevent reasoning deadlocks, we improve MCTS by introducing a new **intervention mechanism for anomaly handling**.

**Key Phases of MCTS.** MCTS-based decision-making in CycleIE consists of four key phases:

*1) Selection*: Starting from the root node $(s, a)$, CycleIE evaluates candidate nodes using the Upper Confidence Bound applied to Trees (UCT), which balances exploration and exploitation:

$$\text{UCT}(v) = \frac{Q(v)}{N(v)} + \alpha \sqrt{\frac{\ln(N(\text{parent}(v)))}{N(v)}} \tag{1}$$

where $N(v)$ tracks the visit count of node $v$ that is initially 0, and $Q(v)$ accumulates the total reward received by node $v$. This allows informed trade-offs between deep analysis of already retrieved documents and broader searches for new information. The UCT score directs simulations toward promising nodes, thereby influencing the visit statistics used in final action selection.

*2) Expansion*: Once a promising node is selected, the system expands potential subsequent actions, filtered through a validity check conditioned on the current state-action pair $(s, a)$:

$$\mathcal{A}_{\text{valid}} = \{a' \mid a' \in \mathcal{T}(a) \wedge \text{Valid}_{\text{ReAct}}(a', s)\} \tag{2}$$

Here, $\text{Valid}_{\text{ReAct}}(a', s)$ evaluates action plausibility within the ReAct framework, ensuring logical coherence (*e.g.,* prohibiting extraction actions before retrieval).

*3) Simulation*: From each newly expanded node, the system simulates action sequences until reaching a terminal state or a predefined simulation depth. Each simulated trajectory is scored by a comprehensive reward function:

$$R(a, s) = R_{\text{base}}(a, s) + R_{\text{verify}}(a, s) - R_{\text{cycle}}(a, s) - R_{\text{penalty}}(a, s) \tag{3}$$

The reward function combines intrinsic action value ($R_{\text{base}}$), verification quality ($R_{\text{verify}}$, measuring completeness, relevance, and accuracy), penalties for repeated actions ($R_{\text{cycle}}$), and task-specific operational costs ($R_{\text{penalty}}$).

*4) Backpropagation*: The simulation reward $R$ is used to update each node along the simulated path, propagating information upward:

$$N(v) \leftarrow N(v) + 1, \quad Q(v) \leftarrow Q(v) + R \tag{4}$$

**Best Action Selection.** After simulations, the system selects the optimal next action $a_{\text{next}}$ based on node visit frequency:

$$a_{\text{next}} = \underset{a \in \mathcal{A}(s)}{\arg\max} \, N(s, a) \tag{5}$$

Here, $\mathcal{A}(s)$ denotes the set of all possible actions from the current state $s$. The system executes the action with the highest visit count—shaped by prior UCT-guided simulations—thereby transitioning to a new state and informing subsequent steps.

**Intervention Mechanism for Anomalies.** While MCTS provides a principled framework for long-term planning, it may still fall into local loops—particularly verification deadlocks—where the system repeatedly cycles through *retrieve–extract–verify* without making progress. To mitigate this, we introduce an 'Intervention Mechanism for Anomalies' that dynamically overrides the action selected by standard MCTS when specific criteria are met:

$$a_{\text{next}} = \begin{cases} \text{reason,} & \text{if } C \wedge V_{\text{passed}} \\ \text{refine,} & \text{if } C \wedge N_{\text{total}} > \mu \wedge N_{\text{refine}} < \eta \\ \text{reason,} & \text{if } C \wedge N_{\text{total}} > \mu \wedge N_{\text{refine}} \geq \eta \\ a_{\text{MCTS}}, & \text{otherwise} \end{cases} \tag{6}$$

Here, $C$ is a binary signal for cycle detection, triggered by repeated verification failures on similar content. $V_{\text{passed}}$ indicates whether any verification has succeeded, while $N_{\text{total}}$ and $N_{\text{refine}}$ track the number of actions and refinements in the current episode. Thresholds $\mu$ and $\eta$ control the allowed action length and refinement depth. When $C$ is active and $N_{\text{total}} > \mu$, the system prioritizes *refine* to avoid redundant attempts. If the refinement budget $\eta$ is also exceeded, it defaults to *reason*ing with current content. This gating mechanism overrides the default MCTS flow when necessary, preventing verification loops and improving robustness in complex cases.

# 4 EXPERIMENT

## 4.1 EXPERIMENTAL SETUP

**Dataset.** This study evaluates various document-based data analysis tasks. We selected **four tasks** (spotlight localization, comparison, clustering, and reasoning chain) and **four document length settings** (ranging from 10k to 250k tokens) from the popular Loong benchmark (Wang et al., 2024b). As the document length increases, the information required to solve the tasks becomes more dispersed.

**Implementation.** For the Agents in the workflow, we use Qwen2-72B-Instruct as the base model, with other settings following the same configuration as Loong and StructRAG (Li et al., 2025). Because the questions in the Loong dataset are complex and require decomposition into multiple steps for analysis, we use a Planner Agent (details in Appendix A) for question decomposition.

**Metrics.** We follow the standard settings of Loong (Li et al., 2024) and StructRAG, using their official codebases for evaluation. Two metrics are used: **LLM score** (0–100), assessing output quality (accuracy, completeness, etc.); and **exact match rate (EM)**, which measures the proportion of perfect LLM scores, serving as an indicator of extraction quality, following the same protocol as Loong.

## 4.2 BASELINES

**LLM-Only (no IE).** We evaluated five advanced long-context LLMs: **(1) GPT-4o**, **(2) Claude3.5-Sonnet**, **(3) Kimi-Chat**, **(4) Qwen2-72B-Instruct**, and **(5) GLM4-9B**.

**One-Pass IE Methods:** We selected four recent, competitive RAG approaches. **While commonly labeled as RAG, these methods operate over provided documents without performing retrieval in our setup, and are thus treated as one-pass extractors: (1) RAG (Lewis et al., 2020)**, which segments the input documents into shorter chunks and employs a retriever to select the most relevant ones as augmentation based on the question; **(2) RQ-RAG (Chan et al., 2024)**, which decomposes and reformulates questions using trained LLMs for better retrieval; **(3) GraphRAG (Edge et al., 2024)**, which builds multi-layer graphs from extracted triples to support structured reasoning; and **(4) StructRAG (Li et al., 2025)**, which extracts and transforms various data types (*e.g.,* chunks, code, tables, graphs) for structured generation. GraphRAG and StructRAG both use Qwen2-72B-Instruct as their base model. For implementation, we adopt the LLM settings from Loong (Wang et al., 2024b) and the RAG settings from StructRAG.

## 4.3 MAIN RESULTS

The main results are presented in Table 2, from which we derive the following key findings:

Table 2: Performance comparison across different models and methods on four sets of Document QA benchmarks. In particular, **Qwen2-72B-Instruct** serves as the base model in our method. We highlight the values corresponding to the base model, the **best** model, and the 2nd-best model.

| Method | Spotlight Locating | | Comparison | | Clustering | | Chain of Reasoning | | Overall | |
|---|---|---|---|---|---|---|---|---|---|---|
| | LLM Score | EM | LLM Score | EM | LLM Score | EM | LLM Score | EM | LLM Score | EM |
| **Set1 (10K-50K)** | | | | | | | | | | |
| GPT-4o | 85.67 | 0.81 | 64.27 | 0.47 | 57.01 | 0.24 | 81.58 | 0.55 | 70.40 | 0.44 |
| Claude3.5-Sonnet | 60.85 | 0.55 | 69.07 | 0.47 | 58.63 | 0.13 | 68.57 | 0.50 | 63.69 | 0.37 |
| Qwen2-72B-Instruct | 68.49 | 0.55 | 60.60 | 0.37 | 47.08 | 0.08 | 70.39 | 0.36 | 60.11 | 0.29 |
| Kimi-Chat | 81.11 | 0.74 | 46.70 | 0.20 | 47.84 | 0.07 | 53.77 | 0.17 | 55.02 | 0.24 |
| GLM4-9B-Chat | 63.11 | 0.53 | 54.10 | 0.27 | 39.50 | 0.08 | 56.32 | 0.28 | 51.43 | 0.25 |
| RAG (Lewis et al., 2020) | 51.08 | 0.35 | 44.53 | 0.27 | 37.96 | 0.05 | 53.95 | 0.35 | 46.11 | 0.23 |
| RQ-RAG (Chan et al., 2024) | 72.31 | 0.54 | 48.16 | 0.05 | 47.44 | 0.07 | 58.96 | 0.25 | 53.51 | 0.17 |
| GraphRAG (Edge et al., 2024) | 31.67 | 0.00 | 27.60 | 0.00 | 40.71 | 0.14 | 54.29 | 0.43 | 40.82 | 0.18 |
| StructRAG (Li et al., 2025) | 74.53 | 0.47 | 75.58 | 0.47 | 65.14 | 0.23 | 67.84 | 0.34 | 69.43 | 0.35 |
| **CycleIE (Ours)** | **91.75** | 0.71 | **79.25** | **0.58** | **77.58** | **0.46** | 72.55 | 0.32 | **78.71** | **0.48** |
| **Set2 (50K-100K)** | | | | | | | | | | |
| GPT-4o | 86.76 | 0.72 | 59.81 | 0.40 | 47.83 | 0.11 | 62.09 | 0.34 | 58.38 | 0.29 |
| Claude3.5-Sonnet | 63.83 | 0.53 | 58.90 | 0.39 | 50.96 | 0.10 | 46.09 | 0.26 | 52.73 | 0.24 |
| Qwen2-72B-Instruct | 64.53 | 0.43 | 42.60 | 0.21 | 38.52 | 0.05 | 51.18 | 0.20 | 45.71 | 0.17 |
| Kimi-Chat | 72.82 | 0.52 | 46.77 | 0.21 | 33.46 | 0.06 | 40.51 | 0.15 | 42.40 | 0.16 |
| GLM4-9B-Chat | 65.04 | 0.54 | 41.80 | 0.23 | 30.72 | 0.02 | 42.34 | 0.17 | 40.19 | 0.17 |
| RAG (Lewis et al., 2020) | 66.27 | 0.46 | 46.28 | 0.31 | 38.95 | 0.05 | 46.15 | 0.22 | 45.42 | 0.19 |
| RQ-RAG (Chan et al., 2024) | 57.35 | 0.35 | 50.83 | 0.16 | 42.85 | 0.03 | 47.60 | 0.10 | 47.09 | 0.10 |
| GraphRAG (Edge et al., 2024) | 24.80 | 0.00 | 14.29 | 0.00 | 37.83 | 0.00 | 46.25 | 0.12 | 33.06 | 0.03 |
| StructRAG (Li et al., 2025) | 68.00 | 0.41 | 63.71 | 0.36 | 46.04 | 0.14 | 54.70 | 0.19 | 60.95 | 0.24 |
| **CycleIE (Ours)** | 86.42 | 0.63 | **78.14** | **0.48** | **72.44** | **0.22** | 63.12 | 0.19 | **72.88** | **0.32** |
| **Set3 (100K-200K)** | | | | | | | | | | |
| GPT-4o | 74.84 | 0.65 | 42.40 | 0.21 | 38.70 | 0.04 | 45.06 | 0.09 | 46.95 | 0.19 |
| Claude3.5-Sonnet | 65.36 | 0.56 | 37.79 | 0.03 | 37.79 | 0.03 | 25.95 | 0.11 | 42.06 | 0.19 |
| Qwen2-72B-Instruct | 46.99 | 0.27 | 37.06 | 0.13 | 31.50 | 0.02 | 35.01 | 0.07 | 35.94 | 0.09 |
| Kimi-Chat | 62.13 | 0.54 | 24.20 | 0.05 | 21.98 | 0.01 | 31.02 | 0.14 | 31.37 | 0.14 |
| GLM4-9B-Chat | 69.19 | 0.56 | 37.99 | 0.18 | 26.63 | 0.01 | 32.30 | 0.09 | 37.36 | 0.16 |
| RAG (Lewis et al., 2020) | 73.69 | 0.55 | 42.20 | 0.27 | 32.78 | 0.02 | 37.65 | 0.13 | 42.60 | 0.18 |
| RQ-RAG (Chan et al., 2024) | 50.50 | 0.13 | 44.62 | 0.00 | 36.98 | 0.00 | 36.79 | 0.07 | 40.93 | 0.05 |
| GraphRAG (Edge et al., 2024) | 15.83 | 0.00 | 27.40 | 0.00 | 42.50 | 0.00 | 43.33 | 0.17 | 33.28 | 0.04 |
| StructRAG (Li et al., 2025) | 68.62 | 0.44 | 57.74 | 0.35 | 58.27 | 0.10 | 49.73 | 0.13 | 57.92 | 0.21 |
| **CycleIE (Ours)** | 83.01 | 0.50 | **75.48** | **0.41** | **72.01** | **0.25** | **62.59** | **0.19** | **72.33** | **0.31** |
| **Set4 (200K-250K)** | | | | | | | | | | |
| GPT-4o | 36.79 | 0.19 | 23.97 | 0.08 | 30.40 | 0.00 | 32.89 | 0.07 | 31.11 | 0.07 |
| Claude3.5-Sonnet | 36.91 | 0.24 | 28.82 | 0.05 | 28.68 | 0.00 | 28.77 | 0.08 | 30.51 | 0.08 |
| Qwen2-72B-Instruct | 33.18 | 0.16 | 26.59 | 0.08 | 29.84 | 0.01 | 25.81 | 0.04 | 28.92 | 0.06 |
| Kimi-Chat | 20.17 | 0.12 | 9.17 | 0.00 | 5.56 | 0.00 | 22.61 | 0.11 | 13.50 | 0.05 |
| GLM4-9B-Chat | 15.67 | 0.12 | 21.33 | 0.05 | 12.35 | 0.05 | 21.04 | 0.05 | 16.84 | 0.05 |
| RAG (Lewis et al., 2020) | 52.17 | 0.24 | 24.60 | 0.10 | 26.78 | 0.00 | 17.79 | 0.00 | 29.29 | 0.07 |
| RQ-RAG (Chan et al., 2024) | 29.17 | 0.08 | 40.36 | 0.00 | 26.92 | 0.00 | 34.69 | 0.00 | 31.91 | 0.01 |
| GraphRAG (Edge et al., 2024) | 17.50 | 0.00 | 26.67 | 0.00 | 20.91 | 0.00 | 33.67 | 0.33 | 23.47 | 0.05 |
| StructRAG (Li et al., 2025) | 56.87 | 0.19 | 55.62 | 0.25 | 56.59 | 0.00 | 35.71 | 0.05 | 51.42 | 0.10 |
| **CycleIE (Ours)** | **83.19** | **0.53** | **68.50** | **0.35** | **67.58** | **0.18** | **50.47** | 0.09 | **66.70** | **0.26** |

**1) Iterative IE vs. No IE.** Built upon the Qwen-72B-instruct model, CycleIE achieves substantial improvements by consistently extracting high-quality structured data across all document length groups, leading to significantly better analysis performance. On the short-document set (Set1), the overall LLM score improves from 60.11 to 78.71, and the EM metric rises from 0.29 to 0.48. The performance gains become even more pronounced as document length increases: 27.17 points on Set2, 36.39 on Set3, and 37.78 on Set4. **These compelling results clearly show that high-quality structured data extraction can significantly enhance end-to-end reasoning accuracy, especially in longer contexts.**

**2) Iterative IE vs. One-Pass IE.** CycleIE consistently outperforms RAG-based baselines across all document length groups. Compared to StructRAG, CycleIE achieves higher scores on Sets 1 through 4 (78.71, 72.88, 72.33, and 66.70, respectively), substantially surpassing StructRAG's scores (69.43, 60.95, 57.92, and 51.42). The performance gap is even wider against RQ-RAG and GraphRAG—for

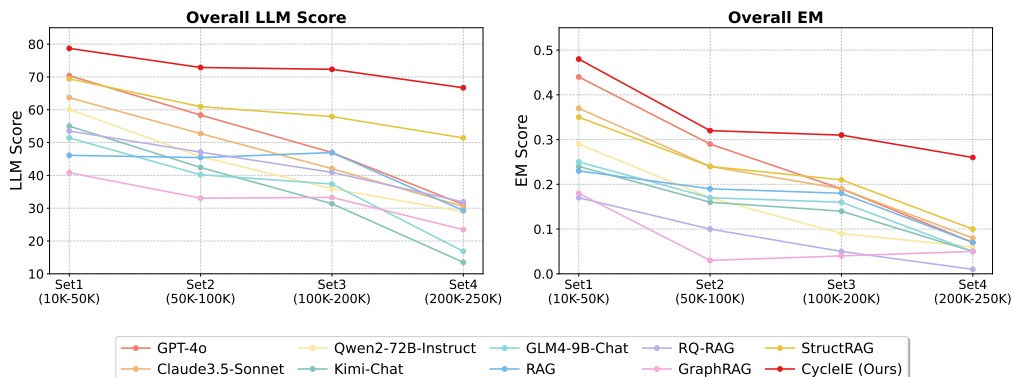

Figure 3: Visualization of overall LLM and EM scores of different methods across sets 1–4.

Table 3: Ablation study.

| | Set 1 | | Set 2 | | Set 3 | | Set 4 | |
|---|---|---|---|---|---|---|---|---|
| | **LLM score** | **EM** | **LLM score** | **EM** | **LLM score** | **EM** | **LLM score** | **EM** |
| **CycleIE** | **78.71** | **0.48** | **72.88** | 0.32 | **72.33** | **0.31** | **66.70** | 0.26 |
| *w/o* verify | 73.75 | 0.43 | 71.40 | **0.36** | 67.12 | 0.30 | 64.16 | **0.27** |
| *w/o* extract | 76.00 | 0.45 | 70.43 | 0.35 | 67.14 | 0.29 | 60.18 | 0.22 |
| LLM-only | 60.11 | 0.29 | 45.71 | 0.17 | 35.94 | 0.09 | 28.92 | 0.06 |

example, on Set3, CycleIE exceeds RQ-RAG by 31.40 points and GraphRAG by 39.27 points. **These results demonstrate that iterative information extraction is more effective than one-pass approaches, enabling better integration of retrieval and reasoning for document-level tasks.**

**3) Performance across Different Task Types.** CycleIE shows strong performance across all task types. On the *Spotlight Locating* task, it consistently achieves high scores from Set1 to Set4 (91.75 to 83.01), significantly outperforming GPT-4o, which drops to 36.79 on Set4. In the *Comparison* task, CycleIE achieves 68.50 on Set4—12.88 points higher than StructRAG. For *Clustering*, it maintains a 12.44-point lead on Set1. Although GPT-4o slightly outperforms CycleIE on the *Chain of Reasoning* task for short documents, CycleIE surpasses it by 17.58 points on Set4. **These results demonstrate that CycleIE outperforms baselines across all sub-tasks.**

**4) Robust Long-Document Handling.** CycleIE is robust against increasing document length. As shown in Figure 3, its overall score decreases by only 15.3% from Set1 (78.71) to Set4 (66.70), compared to much sharper drops for GPT-4o, Claude 3.5 Sonnet, and Qwen-72B (51.9–55.8%), and 25.9% for StructRAG. On Set4, CycleIE outperforms StructRAG and GPT-4o by 15.28 and 35.59 points, respectively. **These results underscore CycleIE 's particular strength in handling long-document inputs, maintaining high reasoning accuracy even as context length increases.**

## 4.4 ABLATION STUDY

To illustrate how each component contributes to our multi-agent workflow, we present representative success and failure cases of CycleIE.

**Criticality of the Verification Module and Inter-Module Synergy.** The verification module is essential for maintaining answer quality. On Set1, its removal (*w/o verify*, 73.75) caused a larger performance drop than removing the extraction module (*w/o extract*, 76.00), highlighting its role in filtering noisy information. Although ablated systems still outperform the base model (*e.g.,*, 67.12 by *w/o verify* on Set3 compared to 35.94 from Qwen2-72B), only the full system consistently achieves the highest scores. On Set2, it reached 72.88, exceeding both *w/o verify* (71.40) and *w/o extract* (70.43), demonstrating the importance of synergy between modules.

**Amplified Benefits of Modularity in Long-Document Processing.** The benefits of modular design grow with document length. From Set1 to Set4, the performance gap between the full system and *w/o extract* widened from 2.71 to 6.52. The impact of removing the verification module also increased with length, causing a 5.21-point drop on Set3 versus 1.48 on Set2. These trends suggest both extraction and verification are increasingly critical for robust reasoning under long-context settings.

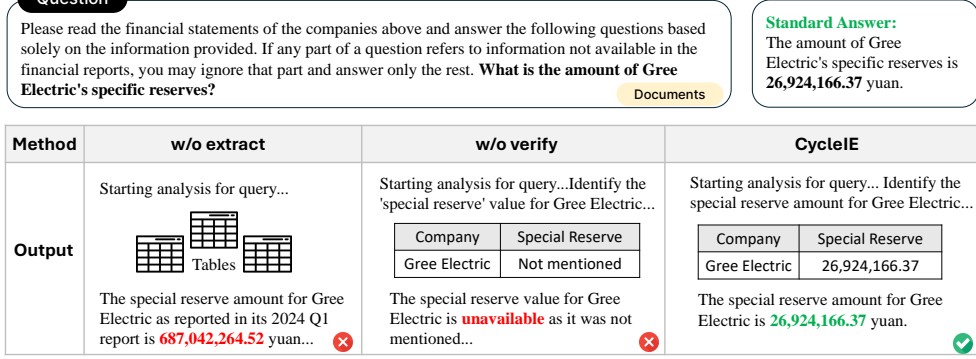

Figure 4: A real running example of CycleIE and its variants from the Loong dataset. Detailed results are provided in the Appendix C.

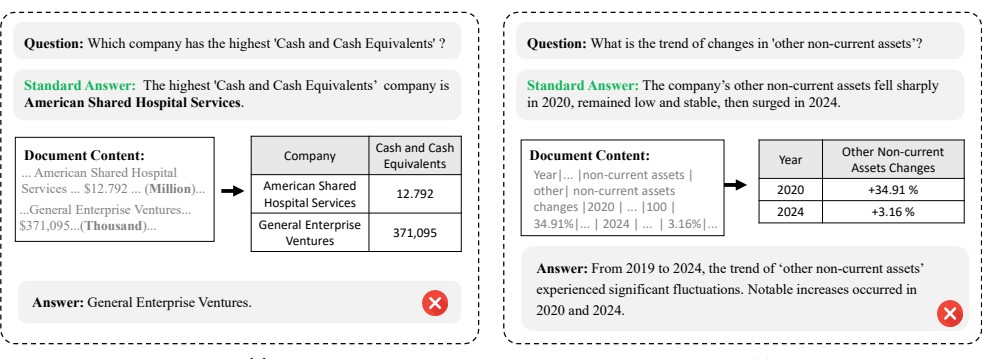

Figure 5: Two failure cases of CycleIE from Loong dataset.

## 4.5 CASE STUDY

This section summarizes representative success and failure cases of CycleIE, showing how its extraction–verification interplay enables robustness in noisy financial disclosures, as well as how early extraction errors can propagate when structural or unit-level inconsistencies are present.

**Successful Cases.** Figure 4 shows why both targeted extraction and reasoning-aware verification are essential for financial QA. The *w/o extract* variant, forced to read raw tables, is overwhelmed by noise and picks an unrelated value, while the *w/o verify* variant finds partial evidence but cannot assess completeness and wrongly concludes the value is missing. In contrast, CycleIE merges both strengths: the extractor isolates focused candidate values, filtering distracting content, and the verifier checks their consistency with the query and context. This coordinated process lets CycleIE correctly identify Gree Electric's reserve amount, showing that extraction or verification alone is insufficient without their interaction in noisy or irregular disclosures.

**Failure Cases.** Despite these strengths, CycleIE also fails when early extraction errors cannot be fixed, as shown in Fig. 5. (a) *Heterogeneous units:* When tables mix units (e.g., millions vs. thousands) and global normalization is absent, the extractor may select the numerically largest token instead of the true value; refinement then reuses this mis-scaled evidence, and the verifier, focused on internal consistency, misses the mismatch. (b) *Table misalignment:* Multi-row headers or irregular formatting can cause misalignment during text conversion, creating incorrect associations (e.g., pairing percentage changes with wrong years). Refinement reinforces these errors, and the verifier, lacking structural cues, may accept the faulty reasoning.

## 4.6 EFFICIENCY AND LATENCY ANALYSIS

To assess the practical deployment feasibility of CycleIE, we analyze its runtime cost and API call frequency compared to direct prompting and human annotation. Table 4 summarizes the average time cost per question.

Table 4: Average runtime per question under different settings.

| Setting | Avg. API Calls | Avg. Time (s) | Notes |
|---|---|---|---|
| CycleIE (ours) | ∼12 | ∼48 | 4s per call to Qwen2-72B-Instruct |
| Qwen2-72B-Instruct (direct) | 1 | ∼4 | One-shot QA |
| Human Annotator | – | 180–600 | 3–10 mins depending on doc size |

Table 5: Answer quality comparison between direct Qwen2-72B-Instruct (LLM-only) and CycleIE across document sets of Loong dataset.

| Set | LLM Score | | EM | | Improvement Ratio | |
|---|---|---|---|---|---|---|
| | LLM-Only | CycleIE | LLM-Only | CycleIE | LLM Score | EM |
| Set1 | 60.11 | 78.71 | 0.29 | 0.48 | 1.31× | 1.66× |
| Set2 | 45.71 | 72.88 | 0.17 | 0.31 | 1.59× | 1.82× |
| Set3 | 35.94 | 72.33 | 0.09 | 0.31 | 2.01× | 3.44× |
| Set4 | 28.92 | 66.70 | 0.06 | 0.26 | 2.31× | 4.33× |

**Computational Cost vs. Performance.** Our system typically decomposes a complex query into approximately 2.5 unit questions, triggering a chain of retrieval, extraction, and reasoning actions. This process results in an average of 12 API calls to Qwen2-72B-Instruct per query. With an average latency of 4 seconds per call, the total end-to-end latency is roughly 48 seconds. While this introduces a significant overhead compared to the ∼4 seconds required for standard one-shot prompting, this additional compute is essential for the performance gains. As shown in Table 5, the extra inference time yields substantial improvements in answer quality, with EM scores improving by up to 4.33× on complex document sets (e.g., Set4).

**User Experience and Scalability.** To mitigate the perceived latency, CycleIE employs real-time streaming, displaying intermediate reasoning steps to the user instantly rather than blocking until completion. Furthermore, compared to human experts—who require 3–10 minutes per question depending on document length—CycleIE offers a highly scalable solution that is orders of magnitude faster, striking a practical balance between automation efficiency and expert-level accuracy.

## 5 RELATED WORK

Document-based information extraction (IE) has evolved from early structured QA systems (Herzig et al., 2020; Eisenschlos et al., 2020) to layout-aware and instruction-tuned models (Appalaraju et al., 2021; Wu et al., 2023b; Feng et al., 2023a; Huang et al., 2022), supporting more flexible extraction across diverse document types. Recent work emphasizes structure-aware modeling (Wang et al., 2024a; Feng et al., 2023b; Masry & Hajian, 2024) to better handle complex layouts and long-range dependencies (Edge et al., 2024; Zhang et al., 2024b). However, most existing methods follow a one-pass extraction paradigm, lacking mechanisms for verification or refinement—often leading to incomplete or inconsistent outputs (Lewis et al., 2020; Li et al., 2022; Huang & Huang, 2024). To address this, recent reasoning frameworks such as ReAct (Yao et al., 2023) and Graph-of-Thoughts (Besta et al., 2024) promote step-wise, feedback-driven reasoning. Adaptive IE methods (Li et al., 2025; Jeong et al., 2024) further explore task-specific structure selection, but still fall short in fine-grained verification. Distinct from prior work, our method introduces a document-specialized multi-agent system (Wu et al., 2023a; Li et al., 2023; Hong et al., 2024; Xie et al., 2023; Wang et al., 2025) that performs iterative extraction and verification, enabling reasoning.

## 6 CONCLUSION

In this work, we introduce CycleIE, an iterative information extraction framework integrating ReAct and MCTS, to address the shortcomings of one-pass extraction in complex documents. Through iterative verification and refinement, CycleIE achieves over a 10% performance uplift on challenging benchmarks, offering a novel approach for obtaining high-quality structured information.

Despite these promising results, CycleIE still has limitations. Its effectiveness may decline when using smaller models as agents in zero-shot settings. In such cases, additional prompt tuning or lightweight fine-tuning may be necessary to maintain robustness and generalization across diverse domains. We leave these directions for future work.

# 7 SUPPLEMENTARY MATERIAL

**Ethics Statement.** We use only public datasets without sensitive data. Though CycleIE reduces some LLM issues through verification, it may still carry biases or errors, so human oversight is important in critical scenarios.

**Reproducibility Statement.** We have made significant efforts to ensure the reproducibility of our work. The full implementation of our proposed method, including model training, evaluation scripts, and instructions for reproducing the reported results, is publicly available at `https://anonymous.4open.science/r/CycleIE`. All experiments can be reproduced by following the provided scripts with the described hyperparameters. Details of dataset preprocessing and experimental settings are included in the supplementary materials.

**Statement on LLM Usage.** We made limited use of a large language model (LLM) as a writing assist tool. The LLM was applied for tasks such as polishing the language, improving readability, and suggesting alternative phrasings. Its role was restricted to helping with grammar correction and stylistic consistency across sections. All research questions, methodologies, experimental designs, analyses, and conclusions were conceived and executed entirely by the authors. The LLM did not generate or alter any scientific content, nor did it contribute to the novelty or validity of the research findings. The authors take full responsibility for the final text.

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

# A AGENT IMPLEMENTATION DETAILS

## A.1 PLANNER AGENT

The Planner agent is tasked with decomposing complex queries into executable subtasks and replanning when necessary.

**Implementation.** The Planner agent utilizes the `task_decomposer` function to break down complex queries into a series of executable steps. It analyzes the original query, considers available documents, and produces a structured plan with dependencies between steps.

---

**Planner Agent Prompt**

As an expert research assistant, analyze and break down this query into clear executable steps:
ORIGINAL QUERY: "{query}"{document_info}
For each step, provide: 1. A clear description of what needs to be done 2. Whether document retrieval is needed for this step 3. Specific keywords to search for (if document retrieval is needed) 4. Dependencies on previous steps (if any) 5. Use as few steps as possible; ideally, complete it in a single step. If it can be done in one step, there's no need for a second step to summarize the first, and just use the original query.
Important: For each step, consider if it depends on results from previous steps. If it does, list those dependencies.
CONSTRAINT: A step can ONLY depend on steps with LOWER step numbers (i.e., steps that come before it). For example, Step 3 can depend on Steps 1 and 2, but not on Steps 4, 5, etc.
The last step must be a synthesis step that integrates results from previous steps to provide the final answer. Also, the last step description should concat the ORIGINAL QUERY in the last sentence.
Output in JSON format with the following structure:

```json
{
    "steps": [
        {
            "step_number": 1,
            "description": "...",
            "requires_document": true/false,
            "search_keywords": ["keyword1", "keyword2", ...],
            "reasoning": "...",
            "depends_on_steps": [list of step numbers depends on]
        },
        ...
    ]
}
```

---

## A.2 RETRIEVER AGENT

The Retriever agent is responsible for retrieving relevant documents or document chunks based on the current task requirements. It features two retrieval modes. The first uses vector retrieval: user input documents are encoded into vectors and stored in a vector database. The second employs Long Context LLM retrieval, leveraging prompts to guide the LLM. The second mode is the default.

**Implementation.** The Retriever agent employs the `context_retriever` function to search for and retrieve relevant document contexts. It leverages a document manager to perform semantic search based on keywords and task descriptions.

---

**Retriever Agent Prompt**

- Each fragment must: - Retain complete semantic meaning (*e.g.,* full paragraphs, full tables, bullet lists, or code blocks). - Preserve structured content (do not cut off tables, item lists, or inline references). - Come from a document that clearly contains information relevant to the question.
- Limitations: - Each fragment should be no longer than 500 tokens (truncate carefully if necessary, without breaking structure). - Return at most 3 fragments, prioritized by most relevant first. - If no content is relevant, return nothing.
—
Output format (one per line, sorted by relevance):

```
<filename>: <relevant content>
<filename>: <relevant content>
...
```

- Do not summarize, generate answers, or include explanations.
- Only output the selected fragments in the format above.
—
Question: {search_query}
Documents: {retrieve_content}

---

## A.3 STRUCTURER AGENT

The Structure Selector agent analyzes retrieved content to identify and determine appropriate structural representations for the information.

**Implementation.** This agent uses the `data_structure_selector` function to decide the most suitable data structure for representing information relevant to the current step (*e.g.,* text, trees, tables, or graphs).

---

**Structure Selector Agent Prompt**

This is a data structure selection task. Based on the given 'question', choose the most suitable data structure to answer the question.
You can choose from the following options:
- Text Description
- Tree
- Table
- Graph
Your answer should be concise and to the point. Return your answer in the following format directly: {answer: data structure}.
The question is: {question}

---

## A.4 EXTRACTOR AGENT

The Extractor agent extracts fine-grained information from documents and transforms raw text into structured data according to the selected structure.

**Implementation.** The Extractor uses the `information_extractor` function to analyze document content and extract relevant information based on the current task requirements and chosen data structure.

---

**Extractor Agent Prompt**

Extract the most relevant information from these document information and transform into the following structure:
STRUCTURE: {structure}
TASK: {step_info['description']}
DOCUMENT INFORMATION: {docs}
EXTRACTION GUIDELINES: 1. Focus specifically on information that directly addresses the task 2. Extract key facts, figures, quotes, and findings 3. Maintain accuracy - don't add information not present in the documents 4. Note any contradictions or uncertainties in the documents 5. Organize the extracted information logically 6. If "STRUCTURE" is a "Graph" or a "Tree," return a tuple of two or three elements.
EXTRACTED INFORMATION:

---

## A.5 VERIFIER

The verifier agent performs verifying over the extracted data.

**Implementation.** The agent uses the `information_verifier` function to verify the information, including scoring, `refinement_suggestions`.

---

**Verify Scoring Prompt**

Verify the following extracted information against the given criteria:
TASK: {step_info['description']}
VERIFICATION CRITERIA: {criteria}
EXTRACTED INFORMATION: {extracted_info}
RETRIEVED CONTEXT: {state.retrieved_context.get(step_num, "No context available")}
Evaluate the information based on these dimensions: - Completeness (1-5): Does it provide all the necessary information for the task? - Relevance (1-5): How relevant is the information to the specific task? - Accuracy (1-5): Based on internal consistency, does the information seem accurate? - Does the information need refinement? (Yes/No) - Is the issue with the extracted information or with the retrieved context? - What specific improvements are needed?
Respond in JSON format:

```
{
    "verification_passed": true/false,
    "completeness": 1-5,
    "relevance": 1-5,
    "accuracy": 1-5,
    "needs_refinement": true/false,
    "issue_source": "extraction" or "retrieval" or "none",
    "refinement_suggestions": "Specific suggestions for improvement"
    ...
}
```

---

## A.6   REFINER

The Refiner agent performs optimizing the unit question.

**Implementation.**   The agent uses the `step_optimize` function to improve the current step through rewording, keyword optimization, and reasoning adjustment. If refinement is insufficient, it can split the step into multiple sub-questions to ensure a clearer and more actionable process.

---

**Refiner Prompt**

You are a task optimization expert, assisting in refining the current step to better answer the question.
MAIN QUESTION: {original_query}
CURRENT STEP BEING PROCESSED: {step_description}
INFORMATION WE HAVE ALREADY EXTRACTED: {extracted_text}
CONTENT WE HAVE RETRIEVED FROM DOCUMENTS: {retrieved_text}
VERIFICATION RESULTS: {verification_text}
First, analyze whether the current step is effective. If it is valid but can be improved, try to:
Improve the step description to make it clearer and more specific.
Optimize the search keywords to make them more targeted.
Adjust the reasoning process and dependencies of the step.
Only consider generating sub-questions if the current step cannot be optimized.
Please return your optimization results in the following JSON format:

```
{
    "steps": [
        {
            "step_number": <step_idx>,
            "description": "Revised step description",
            "requires_document": true/false,
            "search_keywords": ["keyword1", "keyword2", ...],
            "reasoning": "Justification for why this step needs to be optimized this way"
        }
    ],
    "optimization_type": "refine_current_step" or "generate_sub_questions"
}
```

If you choose to generate sub-questions, include 2 to 3 new sub-steps in the `steps` section that will help answer the main question.

---

## A.7   REASONER AGENT

The Reasoner agent performs reasoning over extracted and structured information to synthesize answers or analytical insights.

**Implementation.**   The Reasoner uses various reasoning patterns to draw conclusions from the extracted information.

---

**Reasoner Agent Prompt**

Reason about the following information to address this specific step:
STEP DESCRIPTION: {step_description}
{extracted_content}
{dependencies_text}
Based on this information, provide a clear, well-reasoned answer that addresses the step description. Focus specifically on what this step is asking for, while incorporating any relevant information from dependent steps. Also, the answer should be concise and to the point.
ANSWER:

## B EXPERIMENTAL SETUP DETAILS

### B.1 MODEL AND DATA SETUP

The experiments were conducted on a device equipped with eight NVIDIA 4090 GPUs. During the experimental process, the Qwen2-72b-Instruct model was used. For the Retrieval Agent, in the "LLM-based matching" setup, Qwen-Long was used for retrieval of ultra-long texts. To eliminate the interference caused by differences in retrieval results, we used the retrieval results from StructRAG Li et al. (2025). If, during the verification process, the retrieved content does not meet the requirements, we switch to the original document from Loong Wang et al. (2024b). For the Retriever, we apply Qwen-Long to find the information.

### B.2 EVALUATION METRICS

Following the approach of Loong Wang et al. (2024b), we use GPT-4 as a judge to evaluate the model's output by comparing it against the golden answer and the question's requirements. The evaluation considers three aspects: Accuracy, Hallucinations, and Completeness, with scores ranging from 0 to 100. Two key indicators are used: (1) LLM Scores: the score given by GPT-4, and (2) EM: the percentage of cases that received a score of 100 out of the total number of cases.

---

**LLM Score Generation Prompt**

[Gold Answer]
{std_ans}
[The Start of Assistant's Predicted Answer]
{ans}
[The End of Assistant's Predicted Answer]
[System] We would like to request your feedback on the performance of the AI assistant in response to the user question displayed above according to the gold answer. Please use the following listed aspects and their descriptions as evaluation criteria: - Accuracy and Hallucinations: The assistant's answer is semantically consistent with the gold answer; The numerical value and order need to be accurate, and there should be no hallucinations. - Completeness: Referring to the reference answers, the assistant's answer should contain all the key points needed to answer the user's question; further elaboration on these key points can be omitted. Please rate whether this answer is suitable for the question. Please note that the gold answer can be considered as a correct answer to the question. The assistant receives an overall score on a scale of 1 to 100, where a higher score indicates better overall performance.Please note that if the assistant's answer and the gold answer fully meet the above criteria, its overall rating should be the full marks (100). Please first provide a comprehensive explanation of your evaluation, avoiding any potential bias.Then, output a line indicating the score of the Assistant. PLEASE OUTPUT WITH THE FOLLOWING FORMAT, WHERE THE SCORE IS A SCALE OF 1 TO 100 BY STRICTLY FOLLOWING THIS FORMAT: "[[score]]", FOR EXAMPLE "Rating: [[100]]":
<Start Output>
Evaluation evidence: your evluation explanation here, no more than 100 words Rating: [[score]]
<End Output>
Now, start your evaluation:

---

### B.3 PARAMETER SETTINGS

All controller parameters were fixed across all datasets without tuning. The reward design follows Eq. (3): each rollout starts from a base reward of 0.5, and verifier scores (completeness, relevance, accuracy; each on a 1–5 scale) are normalized and combined using the canonical UCT exploration constant $\alpha = \sqrt{2}$. Penalties for expensive actions (0.2 for `refine`, 0.3 for `replan`) and a cycle penalty of 0.1 per extra attempt (capped at 0.5) are all bounded, ensuring stable exploration. Because verifier feedback is observed at every iteration, the controller automatically retries or reroutes when a step underperforms, making these constants insensitive and not dataset-dependent.

Trajectory and loop constraints serve as simple stability guards rather than tuned hyperparameters. We allow at most $\mu = 6$ retrieval/verification/replan attempts on the same step and at most $\eta = 2$ high-cost refinements (`refine`/`replan`) within a single loop, preventing long or oscillatory reasoning paths.

Verifier thresholds are also fixed across datasets. If the average of completeness and relevance falls below 2 (i.e., $< 40\%$ of the scale), the controller triggers additional retrieval; otherwise it proceeds with refinement and extraction. This effectively requires context quality of at least "3" before the loop settles and worked robustly across all benchmarks.

## C DETAILED OUTPUTS OF CASE STUDY

The detailed outputs of Section 4.5 are as follows. Please note that the original dataset and outputs contain Chinese for ease of reader understanding. For presentation purposes, we have translated them into English without any modifications or polishing.

**1) w/o extract(but with retriever)**

---

**Starting analysis for query:** `{docs}`
Please read the financial reports of the multiple companies above and answer the following questions based only on the content seen above. For parts of the questions that are not covered in the financial reports, you can ignore those parts and only answer the other parts.
**Query:** What is Gree Electric's special reserve?
Loading and processing documents...
**Step 1:** Identify the 'special reserve' value for Gree Electric in its financial statements. This involves searching through the provided documents for Gree Electric's specific financial data.
[TEMP] Retrieving documents related to the query...
[RETRIEVE] Query used: "Identify the 'special reserve' value for Gree Electric in its financial statements. [SEP] Gree Electric special reserve"
[RETRIEVE] Retrieved content:
**"2024 Q1 Report"**: The value of the special reserve is 26,924,166.37 yuan.

```
|Item|Ending Balance|Opening Balance|
|---|---|---|
|Special Reserve|26,924,166.37|26,969,643.44|
```

**"2024 First Quarter Report"**: The value of the special reserve is 11,770,107.42 yuan.

```
|Item|March 31, 2024|December 31, 2023|
|---|---|---|
|Special Reserve|11,770,107.42|2,020,435.65|
```

**"2024 Q1 Report"**: The value of the special reserve is 687,042,264.52 yuan.

```
|Item|End of this reporting period|End of last year|
|---|---|---|
|Special Reserve|687,042,264.52|557,482,954.85|
```

[TEMP] Making inferences...
**Reasoning Result (Step 1):** The 'special reserve' value for Gree Electric as of the end of the first quarter of 2024 is **687,042,264.52 yuan**, according to the "2024 Q1 Report".
**Step 2:** Synthesize result from Step 1.
[TEMP] Making inferences...
**Final Answer:** The special reserve for Gree Electric in the 2024 Q1 Report is **687,042,264.52 yuan**.

---

**2) w/o verify**

---

**Starting analysis for query:** `{docs}`
Please read the financial reports and answer based only on the content seen above.
**Query:** What is Gree Electric's special reserve?
Loading and processing documents...
**Step 1:** Locate 'special reserve' for Gree Electric from financial reports.
[TEMP] Retrieving documents...
[RETRIEVE] Query used: `"Gree Electric Special Reserve 2024 Q1 Report"`

---

[RETRIEVE]
"2024 Q1 Report": Value is 26,924,166.37 yuan. "2024 Q1 Report": Not mentioned for Gree Electric; Huamei Holdings has no record.
[TEMP] Structuring content...
Structure: Table
[EXTRACT]

```
| Company Name     | Special Reserve                  |
|------------------|----------------------------------|
| Gree Electric    | Not mentioned                    |
| Huamei Holdings  | No record of special reserves    |
```

[TEMP] Making inferences...
**Reasoning Result (Step 1):** 'special reserve' for Gree Electric was not mentioned in the 2024 Q1 report.
**Step 2:** Synthesize the result.
[TEMP] Making inferences...
**Final Answer:** The special reserve value for Gree Electric is **unavailable** as it was not mentioned in the provided report.

### 3) CycleIE

**Starting analysis for query:** `{docs}`
**Query:** What is Gree Electric's special reserve?
**Step 1:** Identify the special reserve amount from the 2024 Q1 report.
[TEMP] Retrieving documents...
[RETRIEVE] "2024 Q1 Report": Special reserve = 26,924,166.37 yuan.
"2024 Q1 Report": Gree Electric's special reserve amount is 26,924,166.37 yuan.
[TEMP] Structuring... Table selected.
[EXTRACT]

```
| Company        | Special Reserve Amount | Report Period |
|----------------|------------------------|---------------|
| Gree Electric  | 26,924,166.37 yuan     | 2024 Q1       |
```

[VERIFY] Verification Passed: Completeness 5/5, Relevance 5/5, Accuracy 5/5
[TEMP] Making inferences...
**Reasoning Result (Step 1):** The special reserve is **26,924,166.37 yuan**.
**Step 2:** Synthesize the final answer.
[TEMP] Making inferences...
**Final Answer:** The special reserve for Gree Electric in 2024 Q1 is **26,924,166.37 yuan**.

## D  TECHNICAL IMPLEMENTATION FOR APPLICATION

### D.1  FRONTEND IMPLEMENTATION

Table 6: Frontend Technology Stack

| Framework / Library | Version | Usage |
|---------------------|---------|-------|
| React | 18.3.1 | UI framework |
| TypeScript | 4.9.5 | Strongly typed JavaScript language |
| Axios | 1.6.2 | HTTP request client |
| React Dropzone | 14.2.3 | File drag-and-drop upload |
| React Markdown | 9.0.1 | Markdown rendering |
| Styled Components | 6.1.1 | CSS-in-JS styling solution |

Table 7: Frontend API Endpoints

| Endpoint | Method | Description | Parameters |
|---|---|---|---|
| /api/upload | POST | Upload file | FormData with file |
| /api/process | POST | Process user query | {query, file_paths, use_all_files, stream} |
| /api/files | GET | Get uploaded file list | None |
| /api/files/delete | POST | Delete file | {filepath} |
| /api/reload-index | POST | Reload document index | None |

## D.2 BACKEND IMPLEMENTATION

Table 8: Backend Technology Stack

| Framework / Library | Usage |
|---|---|
| Flask | Backend web framework |
| Flask-CORS | Cross-origin resource sharing |
| FAISS | High-performance vector store for semantic search |
| Roberta Embeddings | Text embedding model |
| Qwen2-72B-Instruct | Large language model |

Table 9: Vector Database Design

| Component | Description |
|---|---|
| FAISS | High-performance vector database for similarity search |
| DocumentManager | Manages FAISS index creation, loading, and search |
| Document-level Index | Creates dedicated vector index per document |
| Global Index | Unified index across all documents |

Table 10: Backend API Endpoints

| Endpoint | Method | Function | Parameters |
|---|---|---|---|
| /api/upload | POST | Upload file | file |
| /api/process | POST | Process query | {query, file_paths, use_all_files, stream} |
| /api/files | GET | List files | None |
| /api/files/delete | POST | Delete file | {filepath} |
| /api/reload-index | POST | Reload index | None |

