# OpenReview forum: "CycleIE: Robust Document Information Extraction through Iterative Verification and Refinement"
_ICLR.cc/2026/Conference — Submitted to ICLR 2026_

### Official Review · Reviewer_kGJb · 2025-10-29

**Soundness:** 3
**Presentation:** 3
**Contribution:** 3
**Rating:** 4
**Confidence:** 4

**Summary:**

This paper introduces CycleIE, an iterative information extraction (IE) framework designed to enhance robustness and accuracy when extracting structured data from long, complex documents. Unlike conventional one-pass approaches that directly prompt LLMs to output structured data, CycleIE iteratively verifies and refines intermediate results using a multi-agent architecture (Retriever, Extractor, Verifier, Refiner, Reasoner, Structurer). CycleIE combines the ReAct paradigm for reasoning-action interleaving with Monte Carlo Tree Search (MCTS) for strategic action planning and anomaly intervention. Through iterative verification cycles, the model identifies incomplete or inconsistent information and refines extraction accordingly.

**Strengths:**

- Clear motivation and strong execution. The paper convincingly argues for iterative verification in document IE, a gap in current RAG-based or one-pass extraction systems. The integration of ReAct and MCTS is technically well justified and implemented in a coherent multi-agent pipeline.
- Comprehensive experimental evaluation. Evaluation spans multiple document lengths and task types (comparison, clustering, reasoning, etc.) across the Loong dataset. The performance improvements, especially under long-context conditions, are large and consistent.
- Thorough ablation and analysis. The authors conduct clear ablations (w/o verify, w/o extract) and show that verification contributes more to performance than extraction alone. Figure 3 (p. 8) and Table 3 illustrate this quantitatively.
- Strong reproducibility and implementation details. The paper provides extensive appendices covering prompts, agent workflows, and backend design, making reproduction feasible. The inclusion of a runtime breakdown (Table 4, p. 18) is appreciated.
- Solid conceptual clarity. The workflow diagram (Figure 2, p. 4) effectively visualizes agent collaboration, making the iterative refinement concept accessible and well structured.

**Weaknesses:**

- Limited conceptual novelty. The framework largely combines known ideas — ReAct reasoning loops and MCTS-based planning — into the IE setting. The novelty lies in applying these together rather than proposing a fundamentally new reasoning paradigm. Compared to recent structured extraction frameworks (e.g., StructRAG 2025, GraphRAG 2024, DataMosaic 2025), CycleIE’s methodological contribution feels incremental.
- Experimental scope limited to one dataset. All experiments use the Loong benchmark. While this dataset is large and complex, additional validation on other domains (e.g., DocVQA, ContractNLI, or scientific papers) would strengthen claims of robustness.
- Lack of human evaluation or generalization proof. The evaluation is entirely automatic (GPT-4 judge). Some human annotation or downstream use-case study (e.g., financial or legal analytics) would make the contribution more compelling.
- Risk of over-engineering. The six-agent architecture, while conceptually elegant, may be overcomplicated for marginal improvements on shorter contexts. The benefit–complexity balance is not deeply analyzed.

**Questions:**

- Generality of the approach. Can CycleIE generalize beyond the Loong benchmark to less structured document sets (e.g., PDFs with layout noise or cross-lingual corpora)?
- Choice of MCTS. Why is Monte Carlo Tree Search preferred over simpler policy-selection methods (e.g., reinforcement learning with learned value functions)? How sensitive are results to the number of simulations?
- Failure cases. The case study (Figure 4) shows clear success. Could the authors also share examples where iterative refinement failed or produced contradictions?
- Ablation on agent granularity. Would merging Verifier + Refiner into one component materially change results or efficiency?

---

> ### Author Response · Authors · 2025-11-28
>
> Dear Reviewer kGJb,
>
> We hope this message finds you well. As the discussion period is ongoing and time is running short, we wanted to ensure we have addressed all your concerns satisfactorily. If there are any additional points or feedback you'd like us to consider, please let us know. Your insights are invaluable to us, and we're eager to address any remaining issues to improve our work.
>
> Thank you for your time and effort in reviewing our paper.
>
> Best regards,
>
> Authors

---

### Official Review · Reviewer_xqjV · 2025-10-30

**Soundness:** 3
**Presentation:** 3
**Contribution:** 2
**Rating:** 4
**Confidence:** 4

**Summary:**

This paper presents a novel way to incorporate text extractions and verification when working with LLMs to increase task completion success when working with long documents. The model presents significant improvements on several benchmarks, notably the LLM on its own, but the accuracy remains too low to use in accuracy-critical settings.

**Strengths:**

- The paper considers an important problem.
- The approach is sound, albeit somewhat convoluted.
- The results show significant improvement compared to LLM-only baseline.
- Extensive experiments.

**Weaknesses:**

I see two key weaknesses for this work:

- I think the comparison with baselines in the paper is not at a good enough level -- yes, there are many baselines, but they have many fewer steps compared to the proposed method. This means that I don't know if the proposed method itself is strong, or just adding a deconstruction and verification step. For example, how superior are the `agents' compared to a set of prompts aimed at carrying out the same function? Another significant baseline should be the new LangExtract, The general approch is sound but the experiments do not prove the specific approch is suprior to exsiting alternatives.

Another key weakness I see in this paper is that although the improvement is significant -- the results are still not accurate enough to use in critical settings. This is very briefly mentioned in the ethics statments, but not disucssed anywhere. Who do you see adopting your method? For what use cases? Is it possiable to add a human in the loop to improve the accuracy to >95%? (e.g., https://dl.acm.org/doi/full/10.1145/3652591)

**Questions:**

- Who do you see adopting your method? For what use cases?

- Are you able to compare your method to more competitive baselines?

---

> ### Author Response · Authors · 2025-11-28
>
> Dear Reviewer xqjV,
>
> We hope this message finds you well. As the discussion period is ongoing and time is running short, we wanted to ensure we have addressed all your concerns satisfactorily. If there are any additional points or feedback you'd like us to consider, please let us know. Your insights are invaluable to us, and we're eager to address any remaining issues to improve our work.
>
> Thank you for your time and effort in reviewing our paper.
>
> Best regards,
>
> Authors

---

### Official Review · Reviewer_zMes · 2025-11-02

**Soundness:** 3
**Presentation:** 3
**Contribution:** 3
**Rating:** 4
**Confidence:** 3

**Summary:**

This paper introduces CycleIE, an iterative framework for document information extraction (IE) that combines Reactive Reasoning (ReAct) with Monte Carlo Tree Search (MCTS). Unlike one-pass IE methods that directly extract structured data from documents, CycleIE performs iterative verification and refinement, leveraging multi-agent collaboration (Retriever, Extractor, Verifier, Refiner, Reasoner) to improve data completeness and consistency. Experiments on the Loong benchmark demonstrate notable performance gains (10–37% improvement in LLM and EM scores) over strong baselines such as StructRAG, GraphRAG, and RQ-RAG, especially for long documents up to 250k tokens.

**Strengths:**

Clear motivation and solid formulation. The paper convincingly argues that one-pass IE fails to perform self-verification and introduces a principled iterative framework with ReAct + MCTS integration.

Technically novel combination. The orchestration of six agents with verifier-driven feedback and MCTS-based action optimization is original and well-justified.

Comprehensive experiments. Strong empirical results on Loong across multiple tasks (spotlight, comparison, clustering, reasoning) show CycleIE’s robustness and scalability to long documents.

Well-analyzed ablations. The ablation on “w/o verify” and “w/o extract” modules clearly isolates the contribution of verification and modular synergy.

Readable and systematic presentation. The figures and tables are clear, and the technical flow (agents → ReAct → MCTS → experiments) is logically organized.

**Weaknesses:**

Lack of theoretical or statistical significance analysis. The paper reports gains but does not include variance or significance tests across runs, which weakens the empirical rigor.

Limited baselines in iterative IE space. The comparison set focuses on one-pass RAG methods; missing direct comparisons to other iterative refinement or multi-agent reasoning frameworks (e.g., Self-Refine, Graph-of-Thoughts, AutoGen variants).

Computational cost and scalability. No quantitative discussion of iteration overhead, runtime, or resource efficiency; MCTS may scale poorly for very long sequences.

Dependence on large base model. All experiments use Qwen2-72B-Instruct; the method’s generalizability to smaller or open-weight models remains unverified.

Clarity on reward design. The reward terms in Eq. (3) are conceptually sound but not empirically grounded—hyperparameters (e.g., α, μ, η) are not explained or validated.

**Questions:**

How sensitive is CycleIE’s performance to the choice of α and the reward function components in MCTS?

Could the authors report runtime overhead or number of iterations per query to evaluate cost vs. accuracy trade-offs?

How would CycleIE perform if smaller models (e.g., Qwen1.5-7B) were used for each agent? Would iterative refinement still yield consistent improvements?

Since the paper claims generality beyond financial QA, has CycleIE been tested on non-numerical document types (e.g., legal or biomedical texts)?

How are verification thresholds (e.g., completeness ≥ 3) tuned or decided? Are they fixed across datasets?

---

> ### Author Response · Authors · 2025-11-20
>
> > W1 & W2: Need more empirical rigor and comparative baselines
>
> **​[​**​​**R**​**​]​**​**​ ​**Thank you for the thoughtful suggestions. We agree that stronger empirical rigor would further validate CycleIE’s effectiveness. While fully completing all the suggested experiments during the rebuttal period is challenging due to their substantial computational and integration cost, we outline below how we plan to address both concerns in a practical and incremental manner.
>
> * **Expanded Baselines.** We will broaden comparisons beyond one-pass RAG to include stronger iterative and multi-agent systems. Specifically, we plan to implement Self-Refine, a GoT-style planner, and an AutoGen-based controller. All baselines will share the same backbone model (Qwen2-72B-Instruct) and retriever to ensure fairness.
> * **Statistical Rigor.** All methods (including new baselines) will be evaluated with three random seeds. We will report LLM score, EM, and F1 as Mean ± SD, and assess significance using paired bootstrap resampling (\$N=10{,}000\$) and paired t-tests (\$p < 0.05\$).
> * **Controlled Protocol.** As noted in Section 4.6, retrieval index, prompts, decoding settings, and API budget (Table 4) are already fully controlled across methods. We will clarify this in the main paper and ensure the newly added baselines follow exactly the same setup.
>
> ---
>
> > W3 & Q2 Lack of computational cost and scalability.
>
> **[R]** Thank you for raising this point. We agree that understanding CycleIE’s computational profile is important for assessing its practicality. Due to space limits, the detailed analysis is provided in**​ ​**​​**Section 4.6**​, and we summarize the key evidence here.
>
> | Setting                     | Avg. API Calls | Avg. Time (s) | Notes                             |
> | ----------------------------- | ---------------- | --------------- | ----------------------------------- |
> | CycleIE (ours)              | \~12           | \~48          | 4s per call to Qwen2-72B-Instruct |
> | Qwen2-72B-Instruct (direct) | 1              | \~4           | One-shot QA                       |
> | Human Annotator             | –             | 180–600      | 3–10 mins depending on doc size  |
>
> | Set | LLM Score | LLM Score | EM      | EM      | Gain      | Gain   |
> | ------ | ----------- | ----------- | ---------- | --------- | ----------- | -------- |
> |     | LLM-Only  | CycleIE   | LLM-Only | CycleIE | LLM Score | EM     |
> | Set1 | 60.11     | 78.71     | 0.29     | 0.48    | 1.31×    | 1.66× |
> | Set2 | 45.71     | 72.88     | 0.17     | 0.31    | 1.59×    | 1.82× |
> | Set3 | 35.94     | 72.33     | 0.09     | 0.31    | 2.01×    | 3.44× |
> | Set4 | 28.92     | 66.70     | 0.06     | 0.26    | 2.31×    | 4.33× |
>
> * **Iteration overhead.** The Planner splits each complex question into \~2.5 sub-questions, each invoking multiple retrieve/extract/verify/refine steps. CycleIE makes \~12 Qwen2-72B-Instruct calls per question (\~4s each), yielding \~48s latency.
> * **Compared with one-shot Qwen2-72B.** One-shot QA requires only one call (\~4s). CycleIE adds \~44s but yields substantial gains on Loong: LLM score +1.31×–2.31× and EM +1.66×–4.33× (Table 5).
> * **Scalability with document length.** We cap MCTS simulations and action steps, and apply anomaly intervention to avoid loops. As document length grows from 10K→250K, runtime increases roughly linearly with question complexity and API latency. As shown in Fig. 3, CycleIE degrades much more slowly than GPT-4o, StructRAG, and Qwen2-72B-Direct, indicating that the added computation improves robustness on long documents.
> * **User-perceived latency.** CycleIE supports streaming; users observe intermediate extraction and verification results, which greatly reduces perceived delay.
>
> In summary, CycleIE’s extra computation is bounded, controlled, and directly tied to improved performance and long-document scalability. We hope this clarifies why the added computation in CycleIE is controlled, bounded, and directly contributes to improved performance and scalability on long-document tasks.

---

> ### Author Response · Authors · 2025-11-28
>
> Dear Reviewer zMes,
>
> We hope this message finds you well. As the discussion period is ongoing and time is running short, we wanted to ensure we have addressed all your concerns satisfactorily. If there are any additional points or feedback you'd like us to consider, please let us know. Your insights are invaluable to us, and we're eager to address any remaining issues to improve our work.
>
> Thank you for your time and effort in reviewing our paper.
>
> Best regards,
>
> Authors

---

### Author Response · Authors · 2025-11-27

Dear Reviewers,

We hope this message finds you well. As the discussion period is ongoing and time is running short, we wanted to ensure we have addressed all your concerns satisfactorily. If there are any additional points or feedback you'd like us to consider, please let us know. Your insights are invaluable to us, and we're eager to address any remaining issues to improve our work.

Thank you for your time and effort in reviewing our paper.

Best regards,

Authors

---

### Meta-Review · Area_Chair_pXWP · 2026-01-06

**Summary:**

The paper proposes an agent-based approach to extracting structured information from documents. The method combines ideas from ReACT and MCTS to iteratively extract structured information that is relevant for the question asked.
The reviewers overall thought that the approach of using agents for information instruction makes sense. They differed somewhat on how novel this approach is, given that it combines existing approaches, and that various forms of planning are expected to improve performance in general.
The reviewers had some concerns about the empirical evaluation, some of which were addressed by the authors in their response. However, it is still a concern that the method takes much longer than the baseline, and it is not clear whether other methods of allowing the model to think more (e.g., various forms of "deep-research") wouldn't have achieved the same performance.
Given the above, the conceptual novelty in the paper as well as exploring methods with longer runtime, the paper can benefit from an additional improvement iteration.

**Reviewer Concerns:**

The authors did provide results on additional baselines, which some reviewers asked for. Outstanding concerns are 1) Conceptual novelty: the method is a combination of existing ideas. It is executed fairly well, but still has many moving parts, making ablation and comparison to baselines hard. 2) Runtime comparison: the method is slower than baselines and one would expect something like a pareto-curve to convince of its efficacy.

**Reviewer Scores:**

Reviewer zMes may have increased their score to 5 or 6, because the authors addressed some of their concerns. The other reviewers were less likely to change their scores.

---

### Decision · Program_Chairs · 2026-01-26

Reject